**Data Availability Statement:** The do-file (for replication of study analyses) is publicly available on the OSF repository at https://osf.io/afrxk/. The

# Publishing, signaling, social capital, and gender: Determinants of becoming a tenured professor in German political science

**Martin Schröder**[1]*, **Mark Lutter**[2], **Isabel M. Habicht**[1,2]

1 Institute of Sociology, University of Marburg, Marburg, Germany, 2 Institute of Sociology, University of Wuppertal, Wuppertal, Germany

◉ These authors contributed equally to this work.
* martin.schroeder@uni-marburg.de

## Abstract

We apply event history analysis to analyze career and publication data of virtually all political scientists in German university departments, showing that each published refereed journal article increases a political scientist's chance for tenure by 9 percent, while other publications affect the odds for tenure only marginally and in some cases even negatively. Each received award and third party funding increases the odds for tenure by respectively 41 and 26 percent, while international experience, social capital and children hardly have a strong influence. Surprisingly, having degrees from a German university of excellence strongly decreases the odds for tenure. Women with similar credentials have at least 20 percent higher odds to get tenure than men. Our data therefore suggests that the lower factual hiring rates of women are better explained by a leaky pipeline, e.g. women leaving academia, rather than because women are not hired even when they are as productive as men. The article contributes to a better understanding of the role of meritocratic and non-meritocratic factors in achieving highly competitive job positions.

## Introduction

Studies in political science show why women are less frequently parliamentarians [1, 2], party leaders [3], cabinet ministers [4] and heads of state [5]. However, why women are less successful within political science itself is largely unclear. This study contributes to answering this question, by showing which factors correlate with getting a tenured professorship in German political science. This is not only of practical importance for young researchers, but also helps resolving theoretical debates on how science operates.

Robert Merton [6: 270] argued that science should be marked by "universalism", which means that "acceptance or rejection of claims entering the lists of science is not to depend on the personal or social attributes of their protagonist; his race, nationality, religion, class, and personal qualities are as such irrelevant", which means that "careers [must] be open to talents" [6: 272]. Others echo this, arguing that "individual performance alone must be the deciding factor in a person's life chances. Opportunities are said to be equal if gender or social

data underlying the study cannot be made publicly available due to ethical concerns imposed by the ethics committee of the University of Marburg. An anonymized, processed data set which can be used with the do-file to replicate study analyses is available on request from the data management department of the University of Marburg. Interested researchers can send a request to get data access via https://data.uni-marburg.de/handle/dataumr/103. Alternatively, they can contact the data management department at the University of Marburg via data@uni-marburg.de if they have any questions.

**Funding:** The project has been funded by the German Federal Ministry of Education and Research under the codename "LEISTUNGSMESSUNG" (FKZ: 01PU17015A; 01PU17015B) within the funding line "Quantitative research on the science sector."

**Competing interests:** Martin Schröder, Mark Lutter and Isabel Habicht have no competing interests.

background play no role" [7: 223, similarly, cf. 8: 42]. The opposite of this is "particularism", where some groups are favored due to "functionally irrelevant characteristics, such as sex and race, as a basis for making claims and gaining rewards in science" [9: 46].

To show who becomes a tenured professor in German political science, this study draws on a unique dataset of CV and publication data from virtually all academic political scientists in Germany. This circumvents a problem that plagues most labor market and academic labor market studies, which often cannot show whether people are unsuccessful because they do not want or do not get a job. German academia, however, is a strict "up or out system", as German law mandates that researchers can—with very few exceptions—only be employed on fixed term contracts in academia for a maximum of 12 years after having graduated. As tenured professorships are virtually the only non-temporary contracts in German academia, academics either get a professorship or are forced out of the system. Everyone who stays in academia is therefore under the same institutionalized pressure to compete for the few tenured professorships. In the following, we show to which theoretical debates an analysis of this process contributes.

## Factors that influence hiring decisions in academia

**Publications.** Classically, Émile Durkheim [10: 121] claimed that there is only one legitimate way to divide work within modern societies: based on the capacity of individuals to perform what is seen as productive within a domain. Simply put, this implies that work should be done by those who are most capable of doing it. But what capacity is needed in the domain of science? Merton [6: 270] claims that "the institutional goal of science is the extension of certified knowledge." Along these lines, studies on university careers argue that "[w]ithin a research university, the most highly valued activity is contributing to the body of certified knowledge. While teaching and service are also valued, in the absence of research productivity a faculty member's efforts at teaching and service are likely to receive little praise. Consequently, under the norm of universalism, advancement in rank should be most strongly affected by research productivity" [11: 703].

Research productivity is often measured through publications, especially in reputable, peer reviewed journals [12: 84, 13: 473, 14: 2]. This is also the case in political science, where "virtually all institutions value peer-reviewed publications over non-peer-reviewed publications, and more over fewer" [15: 510, 513, 16: 99, 17: 105, 18: 185]. Publications, especially in peer-reviewed journals, should therefore contribute to getting tenure.

**Signaling.** However, hiring takes place under uncertainty. Even the most prolific author may stop publishing after tenure. To reduce uncertainty about their prospects, applicants may signal their potential to hiring committees by passing evaluations, visiting prestigious institutions or acquiring third party funding [cf. 19: 356ff.]. Certified evaluations that signal "readiness" for becoming a professor can be the German "habilitation" or the "junior professorship." A habilitation process, for which researchers author a second monograph and/or a collection of journal articles after their doctoral dissertation, is a procedure for which a faculty committee evaluates a candidate's publications, presentation and external review reports. A positive evaluation brings the "venia legendi", the permission to teach and apply for tenured professorships. The so-called "junior professorship" was introduced as an alternative in 2002. After essentially working as an assistant professor for three years, a committee evaluates a researcher, which—if positive—is equivalent to a habilitation. Since 2002, hiring committees can also consider a candidate's publications as equivalent to a habilitation. Many researchers still write a habilitation however, as this may signal "pre-approval" for a professorship, which reduces uncertainty for the hiring committee [16: 101f., 115].

Preapproval can also be signaled through academic awards, which may be given for and thus certify "state of the art" research or teaching, particular creativity or innovativeness in either domain, as well as service to the profession. Academic awards thereby also signal a candidate's potential to deliver what universities may require. Receiving grants, such as funding from the German research council, similarly shows that external committees have evaluated a candidate's work positively, again reducing uncertainty for the hiring committee. This might unduly advantage men. In a much-cited study, Christine Wenneras and Agnes Wold [20: 342] "found that a female applicant had to be 2.5 times more productive than the average male applicant to receive the same competence score." However, analyzing the same grant-giving body about ten years later did not show that women are judged as less competent [21: 185]. Thus, how grants bestow prestige, how this differs between men and women and how this impacts their careers is unclear from the literature.

Having been at a prestigious academic institution may increase scholarly productivity due to context or peer-group effects of socialization and learning. However, it may also signal preapproval and potential, even if it is unrelated to higher actual productivity. Empirical studies argue that this is the case in the US, where "institutional reputations are far more important in determining present perceptions of departmental rank than are corresponding levels of scholarly productivity" [22: 152, also cf. 23: 785f.]. Gerhards et al. [24: 102] claim that similarly, "the German academic system favors those who have degrees from US universities, simply because they carry greater prestige." While German universities traditionally carried similar prestige, this may have changed, as the German state-funded excellence-initiative endowed some universities with the title "universities of excellence." This led to fear that signaling prestige through one's home institution substitutes actual individual productivity in hiring decisions [25–27: 385].

All of these factors might disadvantage women if their work is less recognized. Some studies argue that women may be seen as "less competent than men, even when women are performing at similar levels to their male colleagues" [17: 116, 20: 342, 28: 280, also cf. 29: 478ff., 485, 30: 60]. Research on the so-called Mathilda-effect [31: 314f.] argues that accumulated advantages are less beneficial for women than for men, partly due to a male-dominated academic culture and other forms of female devaluation. It is therefore important to test whether men generally get hired preferentially compared to women and whether this is due to them having more publications, third party funding and other assets.

**Social capital.** Applicants may not only be hired because of their publications or the potential that they can signal, but also based on their social capital [similarly, cf. 8: 42, 16 97f.]. Understood as "resources which are linked to possession of a durable network of more or less institutionalized relationships" [32: 21], social capital can be an asset by providing a strong professional network. Others stress the "strength of weak ties", of merely knowing someone, rather than having a strong relationship [33, 34]. German political scientists speculate that "you must have at least one friend in a hiring committee and you cannot have an enemy" [16: 102]. Indeed, in French political science, PhD candidates are more successful when having either strong or weak social ties in their committee [35]. The chance of getting shortlisted even doubles when a researcher's former PhD advisor is accidentally part of the committee [36: 71, also cf. 37: 112ff.]. In US sociology, hiring among the most prestigious US universities is largely explicable through the social networks that exist between these institutions [38: 258]. However, for German political science, getting tenure is only weakly related to embeddedness in social networks [16: 115].

If researchers were promoted based on who they know, rather than based on what they do, then this could exclude women [39: 4f., 40] who may lack "access to predominantly male academic networks" [similarly, cf. 17: 115, 28: 278], which "convey critical job-related knowledge"

[29: 477, also cf. 41: 450]. Women are also less likely to accumulate social capital, because their needs for mobility are less prioritized in relationships [42: 407].

**Gender and childcare.**   The preceding sections suggested how women may be disadvantaged, as men benefit more from signaling and social capital. This is a problem, since "[p]ublic trust and confidence in academia rests on its ability to efficiently produce accurate and reliable knowledge, some of which may ultimately inform public debate and national policies. The principle of meritocracy is the best method we know to achieve this, and it has served science very well. To not select and promote the most able individuals (regardless of sex, race, and political views) is, therefore, not only unfair to individual academics but potentially damaging to academia and even to society as a whole" [14: 2]. But are women indeed disadvantaged? The literature shows surprisingly unclear results.

The American Political Science Association surveyed all faculty members in US political science departments and related fields, showing that, descriptively, women are only half as likely to get tenure. However, the survey also showed that women publish less, leaving unclear whether their fewer publications explain why women get hired less, which is compatible with meritocracy, or whether women are disadvantaged regardless of their publications [29: 478ff., 485]. Even in highly egalitarian countries such as Sweden, less representation does not clearly amount to disadvantage in the sense that women do not get posts when performing on par with men [14: 14]. Generally, findings are surprisingly unclear. Early studies found that among psychologists with identical CVs, men are preferred for entry-level positions but not for tenure [43: 526]. Others have shown that women with identical credentials are seen as less competent and consequently less hirable by professors in biology, chemistry and physics [44: 2ff.], while Eaton et al. found women to be judged more critically in physics, but not in biology [45: 136]. Yet others have shown that on average, women in different disciplines are clearly favored over men [46: 6]. However, while a large review of the literature argues that "[s]everal experiments have revealed that both female and male raters downgrade hypothetical job applicants who are female", these studies have dealt with undergraduates, so it remains "unclear whether they generalize to the hiring of tenure track professors"[47: 102]. One explanation for different success is that "the stress of childcare and household responsibilities may be greater for women than for men" so that "for men, having children has a positive effect on promotion, although for women, children have a negative effect" [11: 705, 29: 477, also cf. 48: 105, 49: 183]). However, others find that children do not actually depress the likelihood to get a professorship [also cf. 41: 450, 50: 498].

While the findings of these studies are unclear, they tend to share similar problems. Most select samples rather than using a complete population of scientists. In addition, they tend to show how PhD students or postdocs are evaluated, rather than who actually got tenure. Others sample from those who have a PhD or habilitation in the first place, thus biasing their selection towards those who are academically successful in the first place [16: 102f., 115]. Qualitative studies cannot fill this gap either, as they show whether some women perceive themselves to be disadvantaged, but neither whether this represents a broader population, nor whether it is mirrored by lower actual hiring rates [30: 60]. Political scientists therefore bemoan that claims about discrimination in their discipline are so far largely speculation [17: 116]. We contribute to filling this gap with the following methodology.

## Methods and data

### Data

From December 2018 to December 2019, a trained and supervised team of research assistants coded all CV and publication data from personal and faculty websites of academics with at

least one publication in all political science departments of German universities and two research institutes. We complemented this with an email survey, asking every researcher whether and when they had children. The response rate was 64 percent. We checked all data for outliers and made sure that they are not due to erroneous coding.

The resulting dataset contains 36,875 observations clustered in 1,453 researchers, among which 247 are male and 109 are female tenured political science professors. According to the German statistical office, political science has 250 male and 119 female professors [51: 107], which means we have a virtually complete dataset of German political science, so that confidence intervals around effects can be interpreted as actual variation in German academia, rather than resulting from statistical sampling uncertainty.

## Methods

We use nested Cox regressions [52], which estimate how variables increase or decrease the odds of an event, in this case: tenure. To facilitate interpretation, we use hazard ratios. A hazard ratio of e. g. 1.12 implies that a variable increases the odds to get tenure by twelve percent, while e. g. a hazard ratio of 0.78 means that a variable decreases the chance to get tenure by 22 percent. We use robust and therefore increased standard errors, which account for observations within one person depending on each other [53]. For tied events, we rely on the Efron method [54: 143].

## Variables and modeling strategy

Our dependent variable is the duration from a researcher's first publication, and thus from the moment he or she enters a potential "race for tenure," until either getting tenure or reaching the year 2019. Our data is thus right-censored, which is why we use Cox regressions. We examine what increases or decreases the duration until tenure with the following independent variables.

*Female* is a dummy variable to analyze whether men or women are more likely to get a professorship, before and after accounting for other influences. *Incomplete* is a dummy variable that controls for underreporting, by marking researchers who only show "selected" publications on their website. We show later how this missing data is not a problem in our dataset (see Table A4 in the S1 File). The dummy *Before 2002* accounts for prior time periods, after which the above-mentioned changes were introduced in the German tenure process.

A second model additionally accounts for a researcher's productivity through seven accumulated types of publications. *SSCI journal articles* accumulates publications at each time point in (Social) Science Citation Index (SSCI, SCIE) journals. Since only 6 percent of these articles are ranked in the SCIE and 94 percent in the SSCI, we will use the term "SSCI articles" in the following. As such articles underwent a double blind peer review, they are likely to qualify as the "extension of certified knowledge", which Merton [6: 270] claims is science's core task. We also coded the journal impact factor and weighted articles with it. However, since this did not significantly change the results, we use the unweighted version in the final models. This accords with existing studies, which argue that "[t]here is little evidence that the quality of research, as indicated by citations to the articles or the standing of the journals in which the articles are published, affects promotion" [11: 719].

The variable *Non-SSCI journal articles* similarly accumulates publications in non-SSCI journals. *Monographs* covers all monographs and textbooks. We split this variable into monographs published with "regular" and "highly reputable" publishing houses. Two of the authors coded all publishing houses into these two categories. Intercoder-reliability was .73. We constructed a variable for highly reputable publishers if both researchers blindly agreed on the

high reputation of a publishing house, and a second variable for regular publishing houses if they did not. A list of what publishers are qualified as reputable or regular is in the S1 File (see heading "Reputable and regular/undecided publishers"). We also count the number of *Edited volumes* and *Book chapters*. *Gray literature* counts all remaining publications, including reports, working papers, book reviews, as well as listed but not otherwise published manuscripts. It is important to take up these variables because the existing literature has shown that women tend to publish less than men do [55], especially when competing for tenure [56: 281]. Taking up these variables therefore not only shows how publications are related to getting a professorship. Rather, controlling for publications also shows whether women get hired less because they have fewer publications or whether they even get hired less when having the same publications as men.

We adjusted each publication *p* with *p = 2/(n+1)*, *n* being the number of authors. Being the sole author, therefore, counts as one publication. Being one of two authors as .67, being one of three as .5, and so on. We add 1 and log these variables to account for diminishing marginal returns, as e. g. having published 11 vs. 10 articles should count less than having 3 vs. 2 articles.

While the publication variables show measurable productivity, a third model adds signaling variables that account for career stages, measured as years and years squared after a *Habilitation* and *Junior professorship*. This tests how much more likely researchers are to get hired after each career stage, allowing that after some point, researchers are less likely to get hired with each year. Taking up these variables is important because it allows to compare candidates at similar career stages, which is necessary because women tend to drop out of academia more often than men do due to childbirth and thus do not make it to advanced career stages [15, 49: 183, 57: 148, 58: 4184], also because they worry more often than men that children are incompatible with an academic career [59: 4, 60]. It is also important to control for career stages because some of the existing literature argues that women publish less than men do as they do not reach higher career stages [61]. Thus, controlling for career stages also controls for selective departure from academia, which prevents women to advance towards a tenured professorship. Controlling for career stages additionally controls for resources that may come with advanced career stages. It allows to show, in other words, how likely men and women are to get hired, assuming that they make it to similar pre-tenure career stages in the first place.

International experience may signal a researcher's quality and is measured through *International publications* (in a language other than German), *Months abroad* (at institutions outside of Germany), a *Graduate degree* and *PhD from abroad*. To further measure signaling, we control for accumulated prestige as the share of a researcher's degrees (graduation, PhD, habilitation) from so-called *Universities of excellence*, as ranked by the German "excellence initiative." The excellence initiative supports high-performing universities with additional funding to strengthen their research performance. The initiative aims at creating a set of German universities that are able to compete with the best international universities worldwide, e.g., with Ivy League schools in the US (for more information, see: https://www.dfg.de/en/research_funding/programmes/excellence_initiative/index.html). Academic *Awards* is another important signaling factor, so we count all awards that researchers announce on their websites, such as best paper-, teaching- or other awards. The variable *DFG funding* measures how often a researcher has been funded by the DFG, Germany's main and most prestigious funding agency. We coded this from the Gepris databank, which lists all researchers and projects funded by the DFG (https://gepris.dfg.de/). We only take these variables into account after controlling for publications, so they show the effect of signaling net of measurable productivity.

A fourth model accounts for social capital through three measures. *Mobility* counts all moves to a new institution. We also account for the times a researcher acted as an interim

professor, as well as accumulated co-authors, assuming that each of these variables are related to the size of a professional network. Again, we only control for these variables after accounting for others, thus showing the effect of social capital net of publications, as well as signaling through career stages and accumulated prestige.

Model 5 additionally accounts for the effect of children on men and women, by measuring their presence categorically. Researchers with missing data are coded with a dummy variable to account for non-response bias. This allows to estimate whether children have an impact on getting hired, and whether the effect on any of the previous variables is mediated through parenthood.

Last, Models 6 and 7 test the full model specification separately for men and women, to see whether scholarly productivity, signaling, or social capital affect men and women differently, as suggested above.

## Results

### Descriptive differences at tenure

Table 1 displays descriptives for men and women with complete data who just got tenure (see Table A1 in the S1 File for descriptive data on the entire sample, as well as a discussion of this data). As can be seen, both male and female political science professors in Germany spent about fourteen years after their first publications until they were tenured. When getting tenure, men have published about one SSCI article (or 31 percent) more than women (3.25 vs. 4.26), 35 percent more non-SSCI articles, 24 percent more book chapters and 65 percent more gray literature. Men also take significantly longer to get tenure than women after their habilitation or junior professorship (69 percent longer for habilitation, 89 percent longer for junior professorship), and have received DFG funding more than twice as often. All other differences are not statistically significant at the 5 percent level, except that that we know of 43 percent of men that they have children, but only of 33 percent of all women who just got tenure. Non-response to our children-question is not statistically significant at conventional levels, which suggests no gender-bias in answers about parenthood.

Descriptively, this suggests that men and women needed a similar time to get tenure, but when they do get tenure, men have more publications and DFG funding. As a second descriptive analysis, Fig 1 displays the share of women at each career step.

During their last recorded time point in our dataset, 44 percent of predocs are women. But their share among postdocs is a slightly lower 39 percent, and even only 31 percent among those with a habilitation or junior professorship are women. The female share among professors is as high as in the preceding career stage. This suggests that fewer women get professorship because they quit academia before reaching career stages that typically lead to a professorship. At the same time, the prior descriptive table suggests that among those who do get professorships, women are hired with fewer publications and third-party funding.

### Cox regressions

Table 2 displays the results of multivariate Cox-regressions. Model 1 shows that before accounting for other influences, women have a 6 percent, but statistically non-significantly lower chance to get tenure than men. The model also shows that it was less probable to get a professorship before 2002, and that those with "incomplete" data have a statistically non-significantly higher chance to get a professorship, possibly because tenured professors are more likely to only show selected publications on their websites.

Model 2 adds publication variables. A log increase of SSCI articles multiplies the odds for tenure by 2.22 –a very strong effect. On average, 0 logged publications conform to .02 actual

**Table 1. What characterizes men and women with complete data that just got tenure?**

|  | Mean men | Mean women | Difference | % difference | t-test |
|---|---|---|---|---|---|
| Time since first pub | 14.06 | 14.06 | -0.01 | 0% |  |
| SSCI journal articles | 4.26 | 3.25 | 1.01 | 31% | ** |
| Non-SSCI journal articles | 7.00 | 5.20 | 1.80 | 35% | ** |
| Monographs, reputable | 1.14 | 1.01 | 0.14 | 14% |  |
| Monographs, regular | 0.95 | 0.70 | 0.24 | 35% |  |
| Edited volumes | 1.61 | 1.32 | 0.29 | 22% |  |
| Book chapters | 14.73 | 11.89 | 2.84 | 24% | ** |
| Gray literature | 11.72 | 7.12 | 4.61 | 65% | *** |
| Years since habilitation | 2.12 | 1.26 | 0.86 | 69% | *** |
| Years since junior prof | 1.45 | 0.77 | 0.68 | 89% | ** |
| University of excellence | 0.29 | 0.29 | 0.00 | 0% |  |
| Months abroad | 28.19 | 29.77 | -1.58 | -5% |  |
| Graduated abroad | 0.19 | 0.22 | -0.04 | -17% |  |
| PhD abroad | 0.18 | 0.20 | -0.02 | -11% |  |
| International publications | 13.40 | 11.31 | 2.09 | 18% |  |
| Awards | 0.43 | 0.45 | -0.01 | -3% |  |
| DFG funding | 0.51 | 0.24 | 0.27 | 109% | *** |
| Mobility | 3.01 | 3.26 | -0.24 | -7% |  |
| Interim professor | 1.04 | 1.03 | 0.01 | 1% |  |
| Co-authors | 23.41 | 20.15 | 3.26 | 16% |  |
| Childless | 0.20 | 0.29 | -0.08 | -29% |  |
| Parent | 0.43 | 0.33 | 0.10 | 32% | * |
| No child info | 0.36 | 0.38 | -0.02 | -6% |  |

Notes:

* $p < 0.05$,

** $p < 0.01$,

*** $p < 0.001$.

Data based on 205 men and 94 women at the year of tenure and complete data.

(co-author-adjusted) publications, 1 logged publication conforms to 1.6 publications, 2 logged publications to 5.8 and 3 to 15.5. It is approximately these jumps that each log increase shows. To facilitate interpretation, we have replicated all results with a model that uses linear variables (see Table A2 in the S1 File). This shows that each additional SSCI article increases the odds for tenure by 14 percent. However, that logged SSCI articles are more significantly related to tenure, shows that articles have diminishing returns: each article counts less, the more one already has.

A log increase of monographs with a reputable publisher increases the odds for tenure by 48 percent (plus 22 percent for each book), while odds for tenure actually decrease for every book published with a regular publisher (minus 8 percent per book, but the effect is not statistically significant at any conventional level). Non-SSCI articles increase the chance for tenure by 18 percent at the ten percent significance level (but only 1 percent for every article). Edited volumes increase the chance by 36 percent (8 percent for every edited volume), and book chapters by 23 percent (3 percent for each chapter). With the same publications, women have a 23 percent higher chance to get a professorship, but the relationship is only significant at the ten percent level.

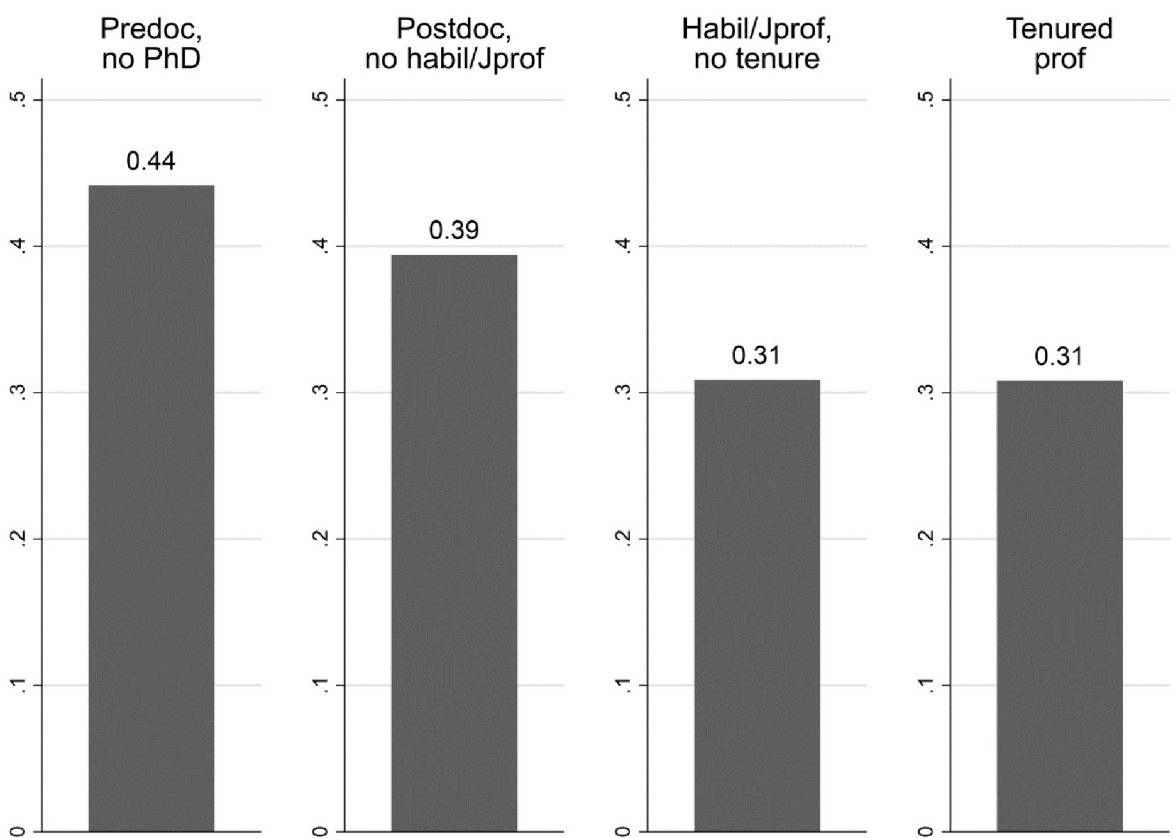

**Fig 1. Share of women at each career step.**

Model 3 adds signaling effects. This renders the gender effect much stronger, implying that women with the same publications and signaling capabilities have 32 percent higher odds to get tenure than men, if only at the five percent significance level. Monographs, even with reputable publishers, lose some of their effect, and edited volumes become not statistically significant at conventional levels, suggesting that they are partially epiphenomenal to signaling: they signal the quality of a researcher, but other signaling mechanisms render their unique influence less important. Having reached advanced career stages also increases the chance for tenure, even net of actual publications. Months abroad increases the chance for tenure by 12 percent, but Table A2 in the S1 File shows no significant linear effect of each month abroad, suggesting that the longer one stays abroad, the less important each additional month becomes. Having graduated abroad has no clear effect, while a foreign PhD increases the chance for tenure. Conversely, researchers whose degrees come from a German "university of excellence" surprisingly have a 33 percent *lower* chance to get tenure. Receiving awards doubles the chance for tenure (plus 45 percent for every award). This suggests that awards strongly signal a researcher's quality, even if they are unaccompanied by publications. DFG funding almost doubles the chance for tenure (with each funding increasing the odds for tenure by 42 percent).

Model 4 adds social capital variables. Because it is the most comprehensive model, Fig 2 visualizes its most relevant effects.

The strongest effect in the full model is that the chance to get a professorship more than doubles with mobility (plus 27 percent for every move, as Table A2 in the S1 File shows).

**Table 2. Main results.**

| | (1) | (2) | (3) | (4) | (5) | (6) | (7) |
|---|---|---|---|---|---|---|---|
| | Gender | Publications | Signaling | Social capital | Children | Women | Men |
| Female | 0.94 | 1.23[+] | 1.32* | 1.20 | | | |
| | (-0.58) | (1.65) | (2.04) | (1.29) | | | |
| SSCI journal articles (ln) | | 2.22*** | 1.74*** | 1.71*** | 1.71*** | 1.84** | 1.64*** |
| | | (9.90) | (5.86) | (5.24) | (5.17) | (2.82) | (4.17) |
| Monographs, reputable (ln) | | 1.48** | 1.22 | 1.04 | 1.03 | 1.03 | 1.05 |
| | | (2.85) | (1.51) | (0.32) | (0.19) | (0.10) | (0.29) |
| Monographs, regular (ln) | | 0.92 | 0.88 | 0.90 | 0.88 | 1.35 | 0.75[+] |
| | | (-0.64) | (-0.94) | (-0.83) | (-0.92) | (0.97) | (-1.90) |
| Non-SSCI journal articles (ln) | | 1.18[+] | 1.19[+] | 1.07 | 1.06 | 1.01 | 1.15 |
| | | (1.93) | (1.88) | (0.76) | (0.64) | (0.03) | (1.26) |
| Edited volumes (ln) | | 1.36* | 1.22 | 1.19 | 1.21 | 0.79 | 1.33[+] |
| | | (2.53) | (1.55) | (1.30) | (1.39) | (-0.75) | (1.77) |
| Book chapters (ln) | | 1.23* | 1.34** | 1.34** | 1.35** | 1.71* | 1.25[+] |
| | | (1.98) | (2.65) | (2.62) | (2.70) | (2.37) | (1.70) |
| Gray literature (ln) | | 1.09 | 1.05 | 1.07 | 1.06 | 1.05 | 1.06 |
| | | (1.34) | (0.66) | (0.93) | (0.85) | (0.35) | (0.69) |
| Years since habilitation | | | 1.65*** | 1.56*** | 1.56*** | 1.51** | 1.70*** |
| | | | (7.75) | (6.47) | (6.53) | (2.71) | (6.71) |
| Years since habilitation$^2$ | | | 0.96*** | 0.96*** | 0.96*** | 0.96* | 0.96*** |
| | | | (-5.23) | (-4.61) | (-4.65) | (-2.06) | (-4.63) |
| Years since junior prof | | | 1.49*** | 1.44*** | 1.46*** | 0.99 | 1.77*** |
| | | | (5.72) | (4.84) | (5.14) | (-0.03) | (6.87) |
| Years since junior prof$^2$ | | | 0.97*** | 0.98** | 0.98** | 1.01 | 0.96*** |
| | | | (-3.57) | (-2.85) | (-3.09) | (0.53) | (-4.41) |
| International publications (ln) | | | 1.00 | 1.01 | 1.01 | 0.93 | 1.07 |
| | | | (0.05) | (0.11) | (0.08) | (-0.44) | (0.59) |
| Months abroad (ln) | | | 1.12** | 1.09[+] | 1.09[+] | 1.11 | 1.10 |
| | | | (2.71) | (1.79) | (1.94) | (1.11) | (1.62) |
| Graduated abroad | | | 0.88 | 1.02 | 1.05 | 1.11 | 1.00 |
| | | | (-0.66) | (0.11) | (0.24) | (0.24) | (-0.02) |
| PhD abroad | | | 1.33 | 1.55* | 1.55* | 1.15 | 2.09** |
| | | | (1.46) | (2.04) | (2.13) | (0.33) | (3.23) |
| University of excellence | | | 0.66** | 0.56*** | 0.53*** | 0.55* | 0.53** |
| | | | (-2.61) | (-3.43) | (-3.69) | (-1.97) | (-3.06) |
| Awards (ln) | | | 2.00*** | 1.90*** | 1.90*** | 2.30** | 1.81** |
| | | | (4.68) | (4.29) | (4.34) | (2.98) | (3.19) |
| DFG funding (ln) | | | 1.84*** | 1.58** | 1.65** | 1.79 | 1.77*** |
| | | | (3.96) | (2.87) | (3.25) | (1.37) | (3.48) |
| Mobility (ln) | | | | 2.27*** | 2.31*** | 2.25*** | 2.51*** |
| | | | | (6.88) | (7.04) | (3.61) | (6.19) |
| Interim professor (ln) | | | | 1.17 | 1.18 | 1.46 | 1.05 |
| | | | | (1.19) | (1.26) | (1.51) | (0.30) |
| Co-authors (ln) | | | | 1.08 | 1.08 | 0.96 | 1.14 |
| | | | | (0.92) | (0.95) | (-0.32) | (1.38) |
| Childless woman | | | | | 1.17 | 1.00 | |
| | | | | | (0.60) | (.) | |

*(Continued)*

**Table 2.** (*Continued*)

| | (1) | (2) | (3) | (4) | (5) | (6) | (7) |
|---|---|---|---|---|---|---|---|
| | Gender | Publications | Signaling | Social capital | Children | Women | Men |
| Father | | | | | 1.32 | | 1.37$^+$ |
| | | | | | (1.52) | | (1.73) |
| Mother | | | | | 1.19 | 1.00 | |
| | | | | | (0.69) | (-0.02) | |
| W/o child info man | | | | | 1.00 | | 1.01 |
| | | | | | (0.00) | | (0.05) |
| W/o child info woman | | | | | 1.66$^*$ | 1.17 | |
| | | | | | (2.30) | (0.51) | |
| Before 2002 | 0.78$^+$ | 0.93 | 1.29$^+$ | 1.35$^*$ | 1.39$^*$ | 1.12 | 1.62$^{**}$ |
| | (-1.85) | (-0.47) | (1.76) | (1.98) | (2.21) | (0.34) | (2.81) |
| Incomplete | 1.27 | 2.22$^{***}$ | 2.22$^{***}$ | 2.27$^{***}$ | 2.23$^{***}$ | 2.45$^*$ | 2.50$^{***}$ |
| | (1.46) | (4.23) | (4.25) | (4.35) | (4.28) | (2.22) | (4.09) |
| $R^2$ | .0014 | .062 | .12 | .13 | .13 | .15 | .17 |
| No. of individuals tenured | 356 | 356 | 356 | 356 | 356 | 109 | 247 |
| No. of individuals total | 1453 | 1453 | 1453 | 1453 | 1453 | 550 | 903 |
| Observations | 35578 | 35578 | 35578 | 35578 | 35578 | 10203 | 25375 |

Exponentiated coefficients; *t* statistics in parentheses; cluster-robust standard errors;

$^+$ $p < 0.1$,

$^*$ $p < 0.05$,

$^{**}$ $p < 0.01$,

$^{***}$ $p < 0.001$.

However, neither having been an interim professor nor having more co-authors is related higher odds for tenure at conventional levels of statistical significance. Note that the inclusion of social capital variables lets the female effect lose its statistical significance and renders it substantially weaker. With the same publications, at the same career stage and with the same international experience as well as social capital, women have a 20 percent and not statistically significant higher chances to get tenure. Note also that the confidence interval for the female indicator is 0.91 to 1.58, so while the effect is not statistically significant at conventional levels, its confidence intervals are clearly more on the side of favoring women than not. Note also that the previous model showed that women have a 32 percent higher chance (significant at the .05 level) to get tenure with the same publications and signaling capacity, but before social capital variables were controlled. This suggests that higher or more effective social capital is one reason why women are more successful (if they have the same publications and signaling capacity). However, the effect of other variables does not change strongly due to the influence of social capital, which suggests that the effect of other variables is not based on an accumulation of social capital.

Model 5 adds the effect of children. Relative to a childless man, an otherwise-similar childless woman has a 17 percent higher chance to get a professorship. However, while a father has a 32 percent higher chance to get a professorship, the chance for an otherwise-similar mother is only 19 percent higher. Note, however, that all of these effects are not statistically significant at conventional levels. The real surprise comes with non-respondents. While men who did not respond to our question about children have the same chance to get a professorship as childless men, women who did not respond have a 66 percent higher chance for a professorship. Thus,

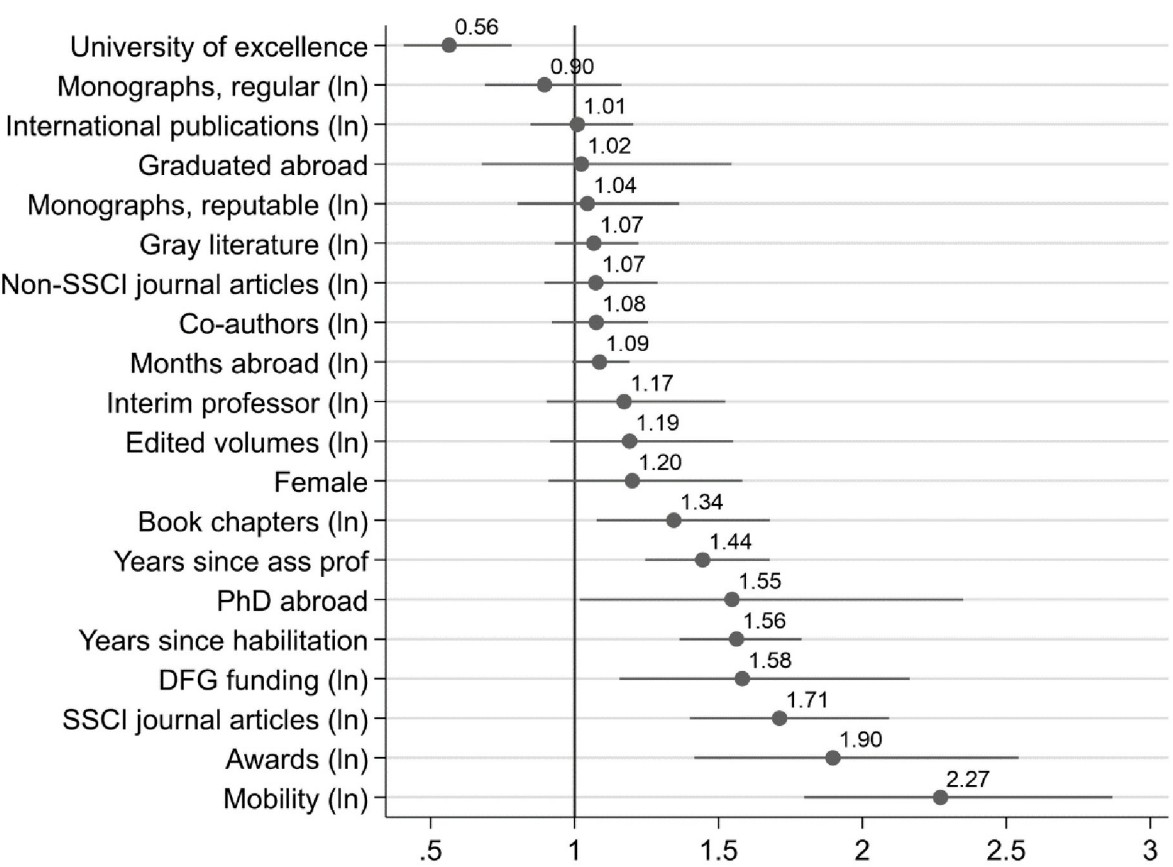

**Fig 2. Effects on chance to get tenure, visualized results based on Model 4 in Table 2.**

women who were more successful than their credentials would suggest were the most reluctant to tell us whether they have children. Importantly, this model also shows that none of the prior effects change after accounting for children. This means that children neither have a strong effect on getting a professorship, nor do they change which variables do have an effect.

Model 6 calculates effects for women and Model 7 for men only. To show which effects influence women significantly differently than men, we also calculated a model where we interact every variable with being a woman (see Table A3 in the S1 File and the visualization in Fig A1 in the S1 File). The comparison shows that women profit twice as much as men from publishing monographs at regular rather than reputable publishing houses. Models 6 and 7 of Table 2 show that this is because men who publish with regular rather than reputable publishers have a lower chance for tenure, while women do not. Also, the chance for women to get tenure increases 1.8-times as much as men's when publishing book chapters, while decreasing when publishing edited volumes, contrary to men's. Also, women get hired less directly after their habilitation. While the effect of other variables also differs between men and women, these others differences are not statistically significant, as Table A3 and Fig A1 in the S1 File, so we refrain from drawing any firm conclusions about them.

## Robustness tests

In separate regressions (see Table A4 in the S1 File), we multiplied each journal article with the impact factor of the publishing journal. Since this data is only available since 1997, we imputed

the 1997 impact factor of 0.4 for prior years. However, weighting each journal article with the impact factor of the journal where it appeared hardly changes the results. This means that while an article in a journal with double the impact factor may indeed count twice as much, our results are not biased because some researchers publish in better journals. We also added the accumulated impact factor of journals where researchers published their articles to the regressions. This had a slightly positive influence, meaning accumulating high impact factor publications is beneficial. However, this effect vanished with controls, which means publishing in highly ranked journals may be a signaling mechanism that can be compensated by other signals, which is similar to what others found [11: 719].

Second, accounting for incomplete data with a dummy variable, as we do in the main calculations, may be problematic. Therefore, in the second model of Table A4 in the S1 File, we only use researchers with complete data. While some results lose significance due to the lower sample sizes, no indicator is strongly affected by incomplete data, which suggests that missing data does not bias our results.

Third, we use a dummy variable to allow for different rules before 2002, the year the German higher education system was reformed. Our results might be different if we exclusively use data from after 2002. This is what Model 3 of Table A4 in the S1 File does. However, results again hardly differ from our main results. In separate calculations, we also interacted every effect with a post 2002 dummy. This shows that monographs with a reputable publisher have become more important and the time after a habilitation less so, but this does not systematically change any of our main conclusions.

Fourth, among those who actually became tenured, different influences may count than among those who did not. Model 4 of Table A4 in the S1 File therefore repeats all calculations with those who did become professors. Results are similar as with the full dataset, which suggests that one important assumption underlying Cox-models is met: Researchers in our dataset are also in our risk-set, as whatever leads to a professorship among those who got a professorship has a similar influence among those who did not. Or to put it differently: what explains who gets a professorship also explains who does not.

Fifth, while a larger share of degrees from a university of excellence may count negatively, it may be important to have studied there, or to have a PhD or a habilitation from there. However, Model 5 in Table A4 in the S1 File shows that having graduated from a university of excellence comes with a slight and statistically not significant advantage, while having a PhD or a habilitation from a university of excellence decreases the hazard for a professorship, but the effect is also not statistically significant. This indicates that having been at a university of excellence indeed does not signal an applicant's quality and, by extension, does not in itself positively influence getting tenure.

Sixth, the children variable may have an impact that varies with the number of children. Notably, the more children a researcher has, the more he or she may be handicapped in getting tenure. We know how many children each researcher has at each point in time, so we calculated—separately for men and women—how a researcher's number of children affects tenure. Compared to men and women without children, fathers have a 69 percent higher chance for tenure with 2 children, and women a 306 percent higher chance with 3 children. All other effects are not statistically significant at the 5 percent level. These results conform to a lifecycle effect, where tenure coincides with the age where people tend to have 2–3 children. The data does not show however, that the chance for tenure systematically declines with the number of children.

Last, some scientists in our dataset may have found one of the very few permanent positions below a professorship, such as lecturer ("Außerplanmäßiger Professor" or "Lehrkraft für besondere Aufgaben"). Such positions pay less and have a higher teaching load. Still,

academics that hold them might not apply for a full professorship, and thus drop out of the "race for tenure." To exclude this as a source of bias, we removed everyone who has an "Außerplanmäßige Professur" as well as those who stayed in our dataset for more than 15 years (and are thus likely on a permanent non-professorial position). This hardly changes our results however, as Table A5 in the S1 File indicates. We therefore conclude that our results are robust to a number of alternative specifications, such as citation-weighting, accounting for missing data, period effects, variable coding decisions and defining risk sets.

## Discussion

On the basis of a multiple source dataset that covers CV and survey data, this paper showed what decreases or increases the duration towards a full professorship. Our results suggest that success in German political science is based on legitimate achievement. Durkheim's [10: 121] idea that hiring decisions should be based on "capacity," and Merton's idea [6: 270], that science is about extending knowledge through "certified scholarship" [11: 703] fits with SSCI articles having the strongest impact on who gets tenure, and with monographs from reputable publishers having a stronger impact than monographs with regular publishers. It confirms what existing studies have suggested, namely that not only quantity, but also measurable quality of publications predicts success in political science [15: 510, 513, 16: 99, 17: 105].

However, signaling factors beyond measurable productivity also have important effects. Notably, having passed formal evaluations through a habilitation or junior professorship increases success irrespective of publications and other observable factors. This is understandable, as having been evaluated from an external committee reduces uncertainty for those who hire an applicant. That awards signal quality may be unsurprising, as they can indicate potential above and beyond what is directly visible through publications. That external funding brings success even in the absence of publications closely mirrors existing findings, such as Mason et al.'s [62: 49] finding that "professors are 65 percent more likely to achieve tenure when directly supported by federal grants." Researchers who bring money may be more desirable candidates because having been chosen by the DFG may signal potential for future research output. However, some may find it worrying that needing more money to do research rather than actually producing more research is in itself a success factor.

A PhD from a foreign university is also a significant effect associated with becoming a professor, which mirrors findings from existing studies [23: 785f., 24: 102]. Discussions on the German excellence initiative expressed fear that German scholars may be recruited based on the prestige of their home institution, rather than based on their actual productivity [25–27: 385]. However, we find no support for this. In fact, we find the opposite. Scholars who passed all their career stages at a German university of excellence have about 40 percent lower odds to get hired; and scholars who have gotten a PhD or a habilitation at a university of excellence also seem to be at a disadvantage. This suggests that international experience is a positive signal, while having been at a German university of excellence is not, which seems counterintuitive and merits further research.

After accounting for productivity as measured through publications and signaling, we find limited effects of social capital on success. Mobility is indeed strongly related to tenure, but this may be because it simply mirrors the willingness to move to a new job. Conversely, acting as an interim professor only has weak effects, just as having more co-authors. This contradicts studies which argue that "who one knows" determines success in academia [35, 36: 71, 38: 258]. It concurs with studies that find rather weak effects [16: 115]. We also did not find that women are disadvantaged because they lack access to (male) social networks [similarly, cf. 17: 115, 28: 278].

 

Existing studies argue that in political science, women are seen as "less competent than men, even when women are performing at similar levels to their male colleagues" [17: 116, 28: 280, also cf. 29: 478ff., 485, 30: 60]. Our results are not compatible with this view. Women with the same publications and resources to signal their quality actually have 32 percent higher odds to get tenure than men. Some of this is due to higher benefits from social capital for women, accounting for which reduces their advantage to 20 percent and to levels that are not statistically significant at conventional levels. We also hardly found that women have to fulfill different criteria than men, contradicting that they are judged by a different standard. If anything, our results indicate that women are judged more benignly than male candidates, which is compatible with some results in political science [16: 115] and beyond [7: 224, 14: 15]. Notably, our results accord with views claiming that "[t]raditionally, observers attempted to explain women's underrepresentation in the academy on the basis of discrimination. We do not deny that women still face discrimination in the academy. However, our findings suggest that traditionally conceived gender discrimination no longer seems to account for the lower rate at which women get tenure-track jobs" [62: 43]. Indeed, if anything, we find that men need to publish more to get hired than women, so that discrimination at the point of hire seems an unlikely candidate to explain the lower proportion of female professors. But if discrimination at the point of hire cannot explain the lower female representation in political science, what can?

Our results fit the "leaky pipeline" hypothesis in political science, which suggests that women get fewer professorships because they are less likely to stay in academia long enough to reach the advanced career stages that lead to a professorship [8: 59, 11, 63: 87]. The data thus indicates that the problem is not that highly qualified women do not get hired when they apply, but that they leave academia before they can apply. Notably, our data shows that with every successive career step, the share of women declines. But among those women who do stay, no discrimination is visible, as the share that makes it from the last career stages to a professorship is as high as among men and at the final stage, women get hired with fewer publications. Thus, efforts to promote female representation in political science should concomitantly focus on why women leave academia, rather than supposing that they are discriminated when applying for tenure, for which we find no evidence.

This confirms some studies, which show that women are hired preferably, compared to similarly qualified men [43: 521, 46: 6]. But it is less compatible with other studies, which find that women are judged as less competent and consequently as less employable in different disciplines [44: 2ff, 45: 136]. Using a review of the existing literature, Ceci et al. [47: 77, 99f., 116] find that though women are underrepresented in very technical fields, they are actually favored when they apply for jobs. However, Ceci et al. also mention that "women are significantly less likely to be promoted in some of the fields in which they are most prevalent: life science and psychology." In this sense, our study might have been a crucial case study [64: 122]. Because compared to STEM fields, we looked at a discipline with a sizeable number of women, where female disadvantage is predicted [47: 116]. Yet we did not find such a female disadvantage. Instead, we found that, if anything, women who apply for tenured professorships in political science have a higher chance of being hired than men with similar qualifications and publications. That women have a higher chance to get hired when they apply may be explained through affirmative action, where one applicant is favored because of their sex. However, affirmative action for women is in fact equivalent to the discrimination of men, who—as we could show—lose out against women, even if they appear more qualified. Some studies have argued that "women must face a choice between having children or succeeding in their scientific careers, while men do not face these same choices" [49: 183, similarly, cf. 57: 148, 58: 4184]. While women with children may leave political science careers more often than men do, we

did not find that having children accounts for women having a lower chance to get hired when they apply.

To sum up, ours is the first study to use a virtually complete sample of all German academic political scientists to show that women tend to be favored over men in the hiring process for tenured professorships, before and after controlling for various factors, most importantly productivity (but see [65, 66] for the discipline of sociology in Germany). This means that women get hired with fewer measurable publications than men do, indicating that there is no bias against women when judging their competency, different from what other studies found [20: 342, 44: 3]. Our study design has a few limitations however. First, we measure productivity through the number of publications. While this accords with existing studies on what should be important in academia, it leaves out teaching. This problem is mitigated because Germany's teaching load is standardized, as a postdoc position (TvöD 13) has four hours of teaching per week, a doctoral student has half that, a junior professor has four hours in the first three, and six hours in the last three years, with minor variation between federal states. Therefore, controlling each person's career steps should be synonymous with controlling teaching load. In addition, empirical studies show that hiring committees usually do not require teaching evaluations [16: 99]. Nonetheless, it would be interesting to see whether quality of and effort for teaching influences who gets tenure. The same is true for impact in terms of citations, which we could model through the impact factor of the journals in which researchers have published. While citations are important, they take time to accumulate, so they may not be adequate to judge non-tenured faculty [14: 3, 16: 100]. Last, while existing studies indicate that administrative experience hardly play a role in who gets tenure [17: 106], a broader definition of productivity, which takes teaching, citations and administration into account, might yield different results. Also, while we could show that women are not disadvantaged in getting a tenured professorship once they have a habilitation or junior professorship, the data indicates that they drop out of academia before they have either. We suspect that women might be disadvantaged in getting a habilitation or junior professorship and that children might be the reason. This however, has to be shown by further research.

## Supporting information

**S1 File. Online annex.**
(DOCX)

## Author Contributions

**Conceptualization:** Martin Schröder, Mark Lutter, Isabel M. Habicht.

**Data curation:** Martin Schröder, Mark Lutter, Isabel M. Habicht.

**Formal analysis:** Martin Schröder, Mark Lutter, Isabel M. Habicht.

**Funding acquisition:** Martin Schröder, Mark Lutter.

**Investigation:** Martin Schröder, Mark Lutter, Isabel M. Habicht.

**Methodology:** Martin Schröder, Mark Lutter, Isabel M. Habicht.

**Project administration:** Martin Schröder, Mark Lutter, Isabel M. Habicht.

**Resources:** Martin Schröder, Mark Lutter.

**Software:** Martin Schröder.

**Supervision:** Martin Schröder, Mark Lutter, Isabel M. Habicht.

**Validation:** Martin Schröder.

**Visualization:** Martin Schröder.

**Writing – original draft:** Martin Schröder.

**Writing – review & editing:** Martin Schröder, Mark Lutter, Isabel M. Habicht.

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
