## [Decision Letter · Decision Letter 0]

28 Oct 2020

PONE-D-20-23645

Publishing, signaling, social capital, and gender: Determinants of becoming a tenured professor in German political science

PLOS ONE

Dear Dr. Schröder,

Thank you for submitting your manuscript to PLOS ONE. After careful consideration, we feel that it has merit but does not fully meet PLOS ONE’s publication criteria as it currently stands. Therefore, we invite you to submit a revised version of the manuscript that addresses the points raised during the review process.

Editor comments

Both reviewers appreciated the extent, depth and quality of the work, and made constructive minor comments that will improve its scholarly value and impact.

As an editor I would add a minor comments about the frequent use of the term "insignificant" for statistical estimates. I would avoid this word, which as the authors surely know is inaccurate, particuarly when the entire population of interest is being measured (although only at a particular time, of course). I would encourage the authors to make greater use of e.g. "not statistically significant" adding where appropriate "at xxx level". Even better, perhaps the authors could consider running a post-hoc power analysis to determine what range of effect magnitudes would fail to reject the null, which would allow them to equate more clearly the failure to reject the null with a "small" (but not necessarily zero or "instignificant") effect.

Reviewers' comments:

Reviewer's Responses to Questions

**Comments to the Author**

1. Is the manuscript technically sound, and do the data support the conclusions?

Reviewer #1: Yes

2. Has the statistical analysis been performed appropriately and rigorously? 

Reviewer #1: Yes

3. Have the authors made all data underlying the findings in their manuscript fully available?

Reviewer #1: No

4. Is the manuscript presented in an intelligible fashion and written in standard English?

Reviewer #1: Yes

5. Review Comments to the Author

Reviewer #1: The authors have done an admirable job collecting and analyzing data on political science in Germany. Their analysis is the most extensive I have seen. And because they are essentially dealing with the entire population of German political scientists, one can have confidence that extraneous factors are not instrumental. I anticipate their findings will be highly cited; I know that my colleagues and I will cite it because it fills gaps that numerous published hiring audits cannot address. There are, however, several minor revisions that these authors could do to strengthen what is already a very strong paper.

First and foremost, they should situate their analysis within the broader international context where the issues of gender bias in academic hiring and promotion are most prevalent. In the present version of the manuscript very few of the major studies are acknowledged. For example, a number of empirical analyses of tenure-track hiring have been reported in the U.S. and Canadian literatures, mostly studies of hiring and promotion at U.S. universities. These studies have been of two types, analysis of gender gaps in actual hiring by universities (I will refer to these as audit studies) and quasi experiments in which faculty are asked to judge the same hypothetical CV for tenure-track hiring or promotion, with a male vs. female name on it. These studies have led to contradictory results. Steinpreiss et al. (1999) found that the hiring of new assistant professors in psychology favored applicants with a male name on the CVs as opposed to the same CV when it had a female name on it. (However, they reported that hiring for more advanced candidates for early tenure did not favor either gender.) Moss-Racusin et al. (2012) found that faculty favored hiring applicants whose CV had a male name over identical female applicants, for baccalaureate-level laboratory manager positions in biology. Easton et al. (2019) reported that faculty favored hiring male postdocs in physics but nonsignificant differences in hiring postdocs in biology. Williams and Ceci (2015) found faculty in four fields favored hiring female applicants for assistant professor positions over male applicants with identical CVs, 66%-33%. In addition, there are numerous studies of the productivity of female applicants for research grants (e.g., Wenneras & Wold, 1997 reported that Swedish female postdocs in biomedicine had to have much greater productivity to get the same grant as a male postdoc received with far fewer publications; however, later, colleagues analyzed the same granting agency and did not find greater productivity by female applicants, in fact, they found that it favored males slightly, Sandstrom & Hallsten, 2007.) By the way, there is an enormous literature on sex differences in productivity, mostly documenting that men publish more at all author positions, in all cohorts, and in most disciplines (e.g., Xie & Shauman,1998; Huang et al. 2020--PNAS). In Huang et al.’s analysis of lifetime publications, political science had a large female advantage in annualized publications and a 3.3% female advantage in total impact. Some contact with this finding is called for as readers familiar with it will want to know why the authors think this occurred.

Obviously, there are big differences in the sampling and analytic approach between these studies and the present German study. It would behoove the authors to acknowledge this larger literature and perhaps attempt to reconcile their contradictory findings. One promising avenue is the authors mention in several places the distinction between applying for jobs vs. being promoted after applying. In the American hiring audits there is a lot of evidence showing that women who earn PhDs are much less likely to apply for tenure-track jobs than are men who earn PhDs.

The presence of children emerges in some past studies (the authors cite Mason et al. but there are many other studies that have found children do not pose a problem for women, once they are hired but their presence does deter women from applying in the first place. These authors’ fourth (or fifth, I forget) model rules this out nicely. And there are analyses of the effect of children as a function of academic discipline and rank (see Ceci, Ginther, Kahn & William’s analyses in the 2014 issue of Psychological Science in the Public Interest.)

Something should be said about the differences in cultural context that may be at work here. On the one hand, Germany has much more extensive child care during the early years than America and perhaps this could be at work. However, the presence of children and the linearity as a function of number of children suggest that childcare is not what is driving the differences between countries, with women who have three children being ~300% more likely to get tenure than childless women, a greater advantage than fathers over childless men. Goulden, Frasch, & Mason (2013, I think) reported that the presence of children or even plans for children in the future were associated with female attrition greater than male attrition among her sample of postdocs. While the authors touch upon gender differences in the willingness to apply for positions, perhaps they could say more as this seems critical to explaining the gender gap they observe. They make a huge contribution in showing that this is not the result of bias against women and that women publish less than men who are hired. Does German academic have something akin to affirmative action like in the U.S.? How else should one explain the seeming preferential tenuring of women with lower productivity? Anyway, there is an enormous literature—far too large to seriously review it—that the authors could selectively cite from to locate the present analysis in this broader context without claiming to have comprehensively reviewed it. However, even without this, the present study is very nicely done and I anticipate that it will be highly cited.

We look forward to receiving your revised manuscript.

Kind regards,

Daniele Fanelli, Ph.D.

Academic Editor

PLOS ONE

Journal Requirements:

Additional Editor Comments (if provided):

Apologies for some delays in completing the reviews, but we have all been working under difficult circumstances.

Reviewers' comments:

Reviewer's Responses to Questions

**Comments to the Author**

1. Is the manuscript technically sound, and do the data support the conclusions?

Reviewer #2: Yes

2. Has the statistical analysis been performed appropriately and rigorously? 

Reviewer #2: Yes

3. Have the authors made all data underlying the findings in their manuscript fully available?

Reviewer #2: No

4. Is the manuscript presented in an intelligible fashion and written in standard English?

Reviewer #2: Yes

5. Review Comments to the Author

Reviewer #2:

Schröder, Lutter and Habicht have done an excellent job on an extremely important topic. One can quarrel with small aspects of their measures and analyses, but these are minor and can be addressed in a revision. One can also quarrel with their failure to cite relevant literature, but again this is a minor omission that can easily be rectified. To me, the importance of their study is that it avoids the common problem of non-representative sampling that limits much of the literature; moreover, it goes beyond the primary question, and they were able to rule out plausible alternative explanations related to productivity and gender bias. I wish these authors were somewhat bolder in their conclusions because they have some of the best evidence that argues against explanations involving bias, showing that if anything female political scientists are actually advantaged in the German system when controlling for various factors, most importantly productivity (i.e., at the professorship stage, women are appointed with fewer publications than men). As these authors point out, this is opposite to past claims of gender bias which argue that women need many more publications than men to achieve the same outcome (e.g., Wenneras & Wold reported this for getting postdoc fellowships in Sweden and many others have claimed it is true for professorial appointments). There are far better cites for this point than the two mentioned. And lest one argue that journals are biased against accepting submissions from female authors (Budden et al. 2008 made this claim, but it has been repeatedly refuted), the evidence when taken as a whole shows this is not true. For example, Berg (2017) reported similar acceptance rates for men and women at the journal Science, as have many others, with only a few exceptions.

There have been many studies of faculty hiring in the U.S. and readers will want to know how these findings compare with the present findings. Williams & Ceci (2015) can be contrasted with Moss-Racusin et al. (2012) and Eaton et al. (2019), although the former dealt with tenure-track hiring of faculty, whereas the second dealt with hiring of lab managers with only baccalaureate training, and Eaton et al. dealt with appointing of postdocs. They differed somewhat in fields (all three included biology but Eaton et al. and perhaps Moss-Racusin et al. may have included physics, and Williams & Ceci included psychology, engineering, and economics as well as biology). And an older study by Steinpreiss et al. (~late 90s) dealt with psychology only. Williams & Ceci dealt with outstanding finalists for tenure track assistant professorships, whereas Eaton et al. dealt with fairly weak postdocs and Moss-Racusin et al. dealt with BA-level lab manager applicants who were described as uneven (strong on some dimensions but unimpressive on others). None of these studies were full population analyses like Schröder, Lutter & Habicht’s. And none were able to rule out various hypotheses like theirs. Thus, the present study has some definite advantages and will be seen as a very important contribution to this literature, which by the way, is one of the most highly cited in all of social science. (One metric is citations: I seem to recall reading that Moss-Racusin et al., which did not deal with faculty hiring and tenure, has been cited around a thousand times as of a few months ago.)

There is also a literature on leakage from the pipeline that the authors might cite. And a large literature on gender gaps in productivity (note--NOT bias in article acceptance, but rather productivity itself, meaning how much work is produced). Basically, all studies find large gender gaps favoring males in amount of work published in journals (again, not bias in acceptance but rather differential output of work with men producing more), regardless of the order of author (first author, corresponding author, any order of authorship), field (some significant differences but all report male advantage), stage of career, nation, etc. Huang et al. (2020) analyzed the full publication history in the Web of Science for over 1.5 million authors and found a large gender gap in favor of men (27% more publishing over a lifetime), but there was no significant gender gap in median publications. Men work about 10% more years and have fewer career interruptions and are three times more likely to be at the extreme right tail of the productivity distribution (Kelchermans and Veugelers, 2013).

A large literature exists in America on the negative role of being a parent when applying for tenure track posts. In addition to the Mason et al. (2013) paper cited, she and Goulden and Frasch as well as Cech & Blair-Loy (2019) and many others have reported that in the U.S., women leave academic careers after childbirth about twice as often as men do following the birth of a child. Many surveys of postdocs and graduate students report that women are much more worried that employment in tenured posts may be incompatible with family life (e.g., Ecklund & Lincoln, 2011; Martinez, 2007). The presence of children or the plans to later have children both reduce women’s likelihood of applying for what Mason and her colleagues call “fast-track” academic careers, i.e., assistant professor posts at research intensive universities. Yet, as far as salary goes, women with young children are quite successful in the NSF Survey of Doctoral Recipients data base, earning higher salaries than those without children.

Minor:

The phrase “lower factual hiring rates of women” is misleading. Women are not actually hired at a lower rate than would be expected based on their representation in the applicant pool; they are underrepresented in the applicant pool, due to earlier leakage in social sciences (but not in physics, economics, CS, mathematics).

---

## [Author Response · Author response to Decision Letter 0]

20 Nov 2020

Please see the uploaded response to reviewers file, where we take into account all suggestions.

---

## [Editor Report · Decision Letter 1]

23 Nov 2020

Publishing, signaling, social capital, and gender: Determinants of becoming a tenured professor in German political science

PONE-D-20-23645R1

Dear Dr. Schröder,

We’re pleased to inform you that your manuscript has been judged scientifically suitable for publication and will be formally accepted for publication once it meets all outstanding technical requirements.

Kind regards,

Daniele Fanelli, Ph.D.

Academic Editor

PLOS ONE
---

## [Editor Report · Acceptance letter]

11 Dec 2020

PONE-D-20-23645R1 

Publishing, signaling, social capital, and gender: Determinants of becoming a tenured professor in German political science 

Dear Dr. Schröder:

I'm pleased to inform you that your manuscript has been deemed suitable for publication in PLOS ONE. Congratulations! Your manuscript is now with our production department. 

Kind regards, 

on behalf of

Dr. Daniele Fanelli 

Academic Editor

PLOS ONE